# Companionship for women/birthing people using antenatal and intrapartum care in England during COVID-19: a mixed-methods analysis of national and organisational responses and perspectives

Gill Thomson ,[1] Marie-Claire Balaam,[2] Rebecca Nowland (Harris) ,[1] Nicola Crossland,[1] Gill Moncrieff,[2] Stephanie Heys,[3] Arni Sarian,[4] Joanne Cull,[1] Anastasia Topalidou ,[2] Soo Downe,[2] ASPIRE-COVID19 Collaborative Group

For numbered affiliations see end of article.

**Correspondence to**
Professor Gill Thomson;
gthomson@uclan.ac.uk and
Mrs Joanne Cull;
Joanne.Cull@nhs.net

## ABSTRACT

**Objectives** To explore stakeholders' and national organisational perspectives on companionship for women/birthing people using antenatal and intrapartum care in England during COVID-19, as part of the Achieving Safe and Personalised maternity care In Response to Epidemics (ASPIRE) COVID-19 UK study.

**Setting** Maternity care provision in England.

**Participants** Interviews were held with 26 national governmental, professional and service-user organisation leads (July–December 2020). Other data included public-facing outputs logged from 25 maternity Trusts (September/October 2020) and data extracted from 78 documents from eight key governmental, professional and service-user organisations that informed national maternity care guidance and policy (February–December 2020).

**Results** Six themes emerged: 'Postcode lottery of care' highlights variations in companionship and visiting practices between trusts/locations, 'Confusion and stress around 'rules'' relates to a lack of and variable information concerning companionship/visiting, 'Unintended consequences' concerns the negative impacts of restricted companionship or visiting on women/birthing people and staff, 'Need for flexibility' highlights concerns about applying companionship and visiting policies irrespective of need, "Acceptable' time for support' highlights variations in when and if companionship was 'allowed' antenatally and intrapartum and 'Loss of human rights for gain in infection control' emphasises how a predominant focus on infection control was at a cost to psychological safety and human rights.

**Conclusions** Policies concerning companionship and visiting have been inconsistently applied within English maternity services during the COVID-19 pandemic. In some cases, policies were not justified by the level of risk, and were applied indiscriminately regardless of need. There is an urgent need to determine how to sensitively and flexibly balance risks and benefits and optimise outcomes during the current and future crisis situations.

## Strengths and limitations of this study

► This is the first paper to consider links between policy and practice in companionship and visiting in maternity care during the COVID-19 pandemic.
► Data triangulation across national level documents, interviews with key stakeholders and public-facing Trust documentation provides nuanced and context-related perspectives on why and how companionship and visiting was impacted.
► Practice-related issues were collected from 25 Trusts websites and social media-based public-facing information, which may or may not reflect actual care practices.
► The paper focuses on antenatal and intrapartum care, with postnatal (including neonatal) care to be the focus of future publications.
► The study does not include information directly reported by parents or healthcare professionals.

## INTRODUCTION

In many cultures around the world, pregnancy is framed as a social event rather than a clinical condition.[1–3] Even where pregnancy, labour and birth are classified as medically risky, social support is expected. Such support, generally referred to as companionship, is usually provided through ongoing family and community relationships, often by female relatives and friends, or community members.[4] As maternity care has become more hospital based, the power to determine who should accompany women/birthing people in clinics and facility settings has shifted to the organisation, and its employees.[5] In the general hospital setting, the public is usually divided into 'patients' or 'visitors'. During the

early decades of mass hospitalisation for antenatal care and birth in the UK, accompaniment for pregnant, child-bearing and postnatal women/birthing people was either disallowed, or conceptualised as 'visiting', and restricted to specific visiting hours. These limitations were justified on the grounds of infection control, overcrowding, privacy for others and defence from potential litigation, if the accompanying companions witness activities they perceive to be negligent or dangerous.[6–8]

Companionship in maternity care is an evidence-based practice with documented benefits in terms of care experiences and clinical outcomes[1] and has been associated with four key attributes: informational support, advocacy, practical support and emotional support.[1] Qualitative studies show that most women/birthing people value companionship, and global guidelines strongly emphasise the need to support and facilitate women's choice to be accompanied throughout the maternity journey.[9] Though restrictions persist in some health economies around the world, companionship and visiting policies in maternity clinics and hospitals in the UK have become increasingly liberal over the last 40 years.

A survey published in 2013 found that over half of fathers/co-parents attended at least one antenatal check, and that 'almost all' were present for ultrasound screening in pregnancy, and for labour.[10] In 2019, 97% of women/birthing people in England said that their partner or someone else close to them was involved as much as they wanted them to be during labour and birth.[11] This is likely related to more inclusive policies as well as consumer demand.

The COVID-19 pandemic has brought the issue of both visitors and companions in health facilities worldwide into sharp focus.[12] In terms of maternity care, there have been anecdotal accounts of wide variations concerning if and whether women/birthing people have been permitted companionship or visitor rights at various points throughout the maternity episode, both between countries, and across different care providers within countries. Concerns about women/birthing people being alone for antenatal contacts, for ultrasound scans (especially when there is bad news), and during labour and birth have been widespread and global in media reports.[12–14]

To understand how companionship and visiting in maternity care during COVID-19 was operationalised organisationally in England during antenatal and intrapartum care, this paper presents an analysis of relevant national level policy and guidance documents, interviews with key stakeholders and a review of public facing information produced by 25 purposively selected maternity providers in England.

## METHODS

This mixed-methods study is part of a larger mixed-methods, observational, multisite comparative research project—Achieving Safe and Personalised maternity care In Response to Epidemics (ASPIRE COVID-19 UK).

Data relating to companionship and visiting during the antenatal and intrapartum period in maternity care were extracted from national policy level documents and interviews with national stakeholders and mapped to analysis of public-facing communication channels from 25 Trusts (maternity care organisations). The Trusts were selected using maximum variation sampling, based on macro-level factors impacting on health inequalities (area-level deprivation reported in the English Indices of Deprivation[15]), meso-level considerations relating to the organisation (Care Quality Commission rating and maternal and neonatal mortality figures) and micro-level factors, including aspects such as parity and access to care identified within national documents.

### Data collection
#### Documentary review
Guidelines, position papers and reports relating to maternity care were collected between February and December 2020 from key governmental, professional and service-user organisations. The organisations were identified as those who provide guidance, campaign and/or advocate for national practice and policy in relation to maternity care. These documents were sourced via organisation-based websites and from key stakeholders involved in the ASPIRE study. All documents that concerned maternity care provision were reviewed with any/all data in relation to companionship and visiting during antenatal and intrapartum care extracted and logged in Excel files.

#### Trust-level public-facing communication about maternity service provision
Data related to companionship and visiting in pregnancy, labour and birth were extracted from Trust websites and Trust-related Twitter, Facebook, and Instagram feeds, between September and October 2020. We extracted information on the format of information presented, access to companionship or visiting antenatally (including ultrasound), and during labour and birth (including induction of labour). We also extracted information that discussed personalisation, organisational response to specific additional needs, Trust response to national guidance, rationale for decisions about companionship/visiting rules and any additional information on companionship/visiting in the context of COVID-19.

#### Interviews
Purposive sampling was used to recruit individuals from relevant national governmental, professional and service-user organisation leads involved in maternity care. Key individuals were identified by project and advisory teams and via snowballing. All participants were approached by email and provided with an information sheet about the study. A consent form was either completed and returned via email, or the consent process was audio recorded at the start of the interview and stored separately from the interview recording. Semi-structured interviews were held July–December 2020 via videoconferencing (using

Microsoft Teams). Interviews lasted between 45 min and 60 min were audio recorded and transcribed in full. The interview schedule (see online supplemental file 1) explored stakeholders' perceptions and experiences of what, why and how changes in maternity care delivery had been made during the pandemic, how changes had been monitored and assessed, and their views on facilitators and barriers to those changes. A total of 50 individuals were approached to participate and interviews were held with 26 stakeholders. While some stakeholders did not respond to the request, others provided names of individuals who they considered would be more suitable. Recruitment was not based on saturation but rather was designed to ensure that we included representation from all key organisations and from individuals that were considered to offer important insights into maternity care delivery.

## Data analysis

All data analysis was undertaken by hand using Excel files. Data analysis for the different forms of data (interviews, documents and Trust-level data) was initially undertaken separately (by RN, M-CB and NC, respectively) before being combined into six key themes using a descriptive content analysis approach[16] led by GT. The stages of analysis were as follows:

► Trust-level data were mapped to different aspects of care (eg, appointments, ultrasound, induction and labour/birth) to identify any variations in maternity service companionship/visiting policies.

► All interview and documentary data that concerned companionship/visiting were extracted and then read on a line-by-line basis to inductively identify meaning units—'the constellation of words or statements that relate to the same central meaning'[17] (p 106). The meaning units were 'manifest' in terms of identifying the visible and salient components in the data.

► The meaning unit labels and associated data were then grouped and synthesised into themes. This process is referred to as 'abstraction' by emphasising descriptions at a 'higher logical level'[17] (p 106);

► In the final phase, the Trust-level data were integrated into the themes for reporting purposes.

Four members of the team (GT, RN, M-CB and NC) were involved in data analysis, and the final themes were agreed by all named authors.

## Public and patient involvement

This study was funded under a rapid response call and while there was no formal public and patient involvement in the original design of the study, UK service-user leads (Maternity Voices Partnership) and members of national charities and from service-user organisations are involved as co-investigators, steering group and advisory members for the ASPIRE project, to ensure that service-user inputs have been considered at every stage of the study.

## Reflexivity

The authors and members of the collaborating group associated with this study are from a range of academic and clinical backgrounds, including midwifery, psychology, obstetrics, neonatology, sociology and social statistics. All the authors are female, and the four interviewers are experienced in undertaking qualitative interviews. One had previously collaborated with some of the interviewees. All authors believe that women/birthing people highly value companionship during key moments in their maternity care, that the priority for policies should be for supporting and facilitating companionship, if desired, and that, for many fathers/co-parents, being present is more than just being a visitor or a supporter. From her psychological background, RN also believes that companions other than fathers/co-parents play a significant role during childbirth in promoting psychological well-being. As midwives, SD, GM, JC and SH view companionship for women/birthing people throughout labour and during antenatal care as a normative practice.

## RESULTS

All the 26 interview participants held a national and/or strategic role in midwifery (n=9), obstetrics (n=1), neonatology (n=1), anaesthesia (n=1), radiography/sonography (n=2) or as an NHS improvement lead (n=1). One was from a doula organisation, three were from the Maternity Voice Partnership (an NHS working group comprising lay members and professionals dedicated to improving maternity care), five were from national charities (focused on birth trauma, premature/sick infants, stillbirth, miscarriage and multiple births) and two service-user organisations that campaign and advocate for maternity care improvements.

All the documents were collected from eight governmental, professional and service-user sources (box 1), with a total of 78 documents providing evidence for the paper (see online supplemental file 2 for full details/references).

The public-facing data logged between September 2020 and October 2020 demonstrated a very wide range of policies and practices between the 25 included Trusts for companionship/visiting during four specific maternity care episodes (antenatal scanning, antenatal

---

**Box 1  Organisations included for documentary analysis**

► Sands (national charity focused on stillbirth).
► AIMS (service-user organisation dedicated to improving maternity care).
► Royal College of Midwives (RCM).
► Royal College of Obstetricians and Gynaecologists (RCOG).
► Society of Radiographers (SoR).
► International Society of Ultrasound in Obstetrics and Gynaecology.
► NHS England (NHSE).
► Birthrights (BR) (service-user organisation dedicated to improving maternity care).

---

appointments, antenatal ward stays and intrapartum) (see table 1). While this could be explained by different COVID-19 infection exposure rates, this may not explain variation between Trusts in the same region.

## Details of themes

Overall, six themes emerged from synthesising the meaning units from the documentary and interview data sets (see table 2). (Further details of the documents and interviews that generated data for each theme are provided in online supplemental file 3.)

### Postcode lottery of care

The notion of a postcode lottery of maternity service provision gained traction in the media over the summer of 2020.[18] Concerns were reflected in national documents from almost all included organisations, the stakeholder interviews, and reflected in Trust level responses. Variation was justified by some organisations as a reaction to local need, for example: 'restrictions on other visitors should follow hospital policy and national guidance' (RCM_7) and 'all staff should work to the same local policy, to provide a consistent service to women' (SoR_5). The caveat to most guidance was that any policies needed to be re-addressed in the event of local spikes in COVID-19 cases or following local risk assessment, to ensure a 'consistent service to women' (SoR_2):

> But it's guidance, and every hospital will make its own decisions and to a certain extent, they will have to because the physicality, the layout, the facilities that they have in hospitals will differ. You know, there's not much room in the waiting area if the corridors are very narrow [so] that the people can't have a two-metre distance. (Stakeholder 20, National charity)

Some Trusts were identified as having 'gone out of their way to ensure their services remain family-centred' (BR_14), whereas in others, partners or other companions of choice were unable to attend any antenatal appointments or scans (BR_16). NHS England released guidance in September 2020 intended to assist Trusts to reintroduce access for companions (NHSE_8). Some organisational responses claimed that this led to 'some Trusts starting to backtrack and reduce restrictions in maternity services' (BR_17), while other Trusts continued to impose restrictions (BR_18) and often without a clear rationale for these variations:

> And that's the problem. You … because down the street, down the road, you could get a very warm, empathetic ultrasonographer who says, of course, yeah, I realise how difficult it is. You know, it doesn't take much, and we just have lost it because people are stressed and there's lots of reasons for it [restricting companions], but it's just not good enough. (Stakeholder 18, Midwifery—strategic role)

### Confusion and stress around 'rules'

Concern over a lack of clarity in decision-making and changes in policies around companionship/visiting were highlighted. For instance, letters from Birthrights to maternity leads (eg, BR_12; BR_16; BR_19) repeatedly emphasised the need for clear reasons, evidence and justification as to why decisions were being made. Concerns included that Trusts 'acted too quickly to withdraw services' and 'decision-making has not always been proportionate or transparent' (BR_18). While most Trusts made some reference to infection risk as the rationale for the restrictions, a sizeable minority (9/25) did not. Many Trusts offered no rationale as to why partners could attend some appointments but not others, for example:

> One birth partner may attend for the 20-week anomaly scan only. Women to attend all other scan appointments unaccompanied. (South-west 2)

Birthrights and some stakeholders highlighted a failure to communicate local restrictions in a timely manner, compounded by rules changing rapidly and difficulties in communicating these changes widely and consistently to large numbers of healthcare professionals. This confusion about the rules was also compound by changeovers of staff and communication between different teams in some areas—'there's been no consultation with sonographers in terms of risk assessments […] or changes in practice' (Stakeholder 26, Radiographer), and services being provided by staff from outside the maternity team.

While some stakeholders noted that individual Trusts responded to this confusion by using a range of public communications, data from the 25 maternity Trusts found that less than half (10/25) had a consistent message about companionship or visiting across different channels. Parent frustration reported by stakeholders also related to how the rules for companions and visiting seemed at odds with the social distancing rules outside of the hospital context:

> In the middle of lockdown, it was 'we don't like it, but we know you're keeping us safe'. Now 'it's we don't like it and I don't see how you're keeping us more safe doing this because I can meet my partner in the pub, but he can't come to my scan. I can do this, but I can't do that'. Yes. So, it's more of an angry mood now. (Stakeholder 12, Maternity Voices Partnership)

In some of the documents by Birthrights (eg, BR_14; BR_16) and the Royal College of Midwives (RCM) (ie, RCM_31) they called for the harm caused by restricted access by companions to be properly and transparently considered within the decision-making processes:

> We would be grateful if you could publish or send us the risk assessment that quantifies the increased risk of spreading COVID-19 within the unit (despite PPE and other mitigating factors, and the fact that most partners are from the same household) caused by relaxing restrictions, and weigh this against the known

**Table 1** Public-facing information on maternity service companionship and visiting policies during antenatal and intrapartum care in 25 English maternity Trusts, September–October 2020

| | Antenatal | | | Intrapartum | | |
|---|---|---|---|---|---|---|
| | **Ultrasound** | **Antenatal appointments** | **Antenatal ward** | **Number of birth partners** | **Timing** | **Induction of labour** |
| Greater London 1 | Unaccompanied | Unaccompanied | Booked time slot | One | Throughout labour | Partner allowed if assessed as needing support |
| Greater London 2 | Partner allowed (20 weeks)* | Unaccompanied | 14:00–18:00 | Two | Not specified | No information |
| Greater London 3 | Partner allowed (20 weeks) | No details | No visitors | One (two if additional need identified) | Established labour | No information |
| Greater London 4 | Unaccompanied | Unaccompanied | No visitors | One | Established labour | No information |
| Greater London 5 | Partner allowed | Unaccompanied | Daytime | One (two if need identified) | Not specified | No information |
| Greater London 6 | Partner allowed | No details | Daytime | One | Throughout labour | Partner allowed daytime only |
| South-east 1 | Unaccompanied | One companion | Booked time slot | One | Not specified | No information |
| South-east 2 | Partner allowed (12 weeks and 20 weeks) | No details | Daytime | Two | Not specified | Partner allowed |
| South-east 3 | Unaccompanied | Unaccompanied | Daytime | One | Throughout | No information |
| South-west 1 | Partner allowed (12 weeks and 20 weeks) | Unaccompanied | Booked time slot | One | Not specified | Daytime only |
| South-west 2 | Partner allowed (20 weeks) | Unaccompanied | No visitors | One | Throughout | Partner not allowed until labour has started |
| South-west 3 | Partner allowed (20 weeks) | No details | No information | One | Throughout | Partner allowed daytime or if additional support needed |
| West Midlands 1 | Unaccompanied | Unaccompanied | No visitors | One | Not specified | No information |
| West Midlands 2 | Unaccompanied | Unaccompanied | No information | One | Not specified | Partner allowed if language/ communication needs |
| East Midlands 1 | Partner allowed (12 weeks and 20 weeks) | Unaccompanied | No information | One | Throughout | No information |
| East of England 1 | Partners allowed | No information | Booked time slot | One | Throughout | No information |
| East of England 2 | Partner allowed (20 weeks) | Unaccompanied | No information | One | Established labour | Partner not allowed until labour has started |
| East of England 3 | Partner allowed (12 weeks and 20 weeks)† | Unaccompanied | 14:00–18:00 | One | Not specified | No information |

Continued

**Table 1** Continued

| | Antenatal | | | Intrapartum | | |
|---|---|---|---|---|---|---|
| | Ultrasound | Antenatal appointments | Antenatal ward | Number of birth partners | Timing | Induction of labour |
| Yorkshire and Humber 1 | Partners allowed | No details | No visitors | One | Not specified | Partner not allowed until labour has started |
| Yorkshire and Humber 2 | Partner allowed (12 weeks) | No details | Booked time slot between 13:00 and 17:00 | One | Throughout | No information |
| North-west 1 | Partner allowed (12 weeks and 20 weeks) | Unaccompanied | Booked time slot | One | Throughout | One partner allowed |
| North-west 2 | Partners allowed | Unaccompanied | No information | Two | Not specified | No information |
| North-west 3 | Partner allowed (20 weeks) | Unaccompanied | No information | Two | Not specified | No information |
| North-east 1 | Partner allowed (12 weeks and 20 weeks)‡ | Unaccompanied | No information | One | Not specified | No information |
| North-east 2 | Partners allowed | No details | No information | One | Not specified | No information |

The term 'partner' is used in this table as this, and 'birth partner', were the most commonly used terms to refer to an antenatal or intrapartum companion.
*Phone call offered if clinical concerns.
†Video offered of a small section at the end of scan.
‡Phone call offered (end of scan).

harms to pregnant women, birthing people and their families from keeping the current restrictions in place. (BR_16)

### Unintended consequences

Almost all organisations, highlighted that having trusted companions throughout labour and birth is linked to improved outcomes, and a lack of companionship was associated with increased need for pharmacological or other interventions (AIMS_8; BR_23). This included perceived impacts on the labour process due to, for example, (increased) 'demand for epidurals' (RCM_8). Alongside the obvious fear and anxiety of infection, organisations and stakeholders highlighted concerns about women/birthing people feeling 'petrified' (Stakeholder 15, Midwifery—strategic role) or not accessing maternity care, 'due to the women's preferred birth partner not being allowed to accompany her' (RCM_2). There were also concerns of companions feeling 'unsupported and uncared for' (Stakeholder 20, National charity) due to being unable to be with the woman/birthing person when they heard bad news (during ultrasound) or missing the birth due to 'being told that they should wait in the car park or something' (Stakeholder 7, Service-user organisation). All but four Trust websites contained messages of empathy regarding the restrictions, sometimes alongside expressions of regret and/or justifications for their necessity: 'We understand the restrictions we have had

in place over recent months have been particularly hard for pregnant women and their families' (North-west 1). Concerns were also expressed by stakeholders and within documents by Sands and Birthrights, towards women/birthing people who had experienced prior baby loss or who may receive bad news alone during the scan:

> Women are being asked to attend scans alone, with many sharing frustrations that they cannot video link to their partners. These very vulnerable women tell us they are concerned about having to attend stressful antenatal appointments and scans alone. While units are being encouraged to consider facilitating women to take a video clip at the end of an appointment, this is reliant on local policies. (Sands_1)

Companionship was noted to have practical as well as emotional implications for women/birthing people. Some of the documents claimed that the absence of companions meant that women/birthing people required more support from maternity care professionals (eg, RCOG/RCM_1j; NHSE_1) creating additional stress for over stretched services (RCOG/RCM_1L) and additional potential exposure to COVID-19 infection. Some practitioner respondents also reported that they or their colleagues experienced moral distress when social distancing rules prohibited physical contact with women/birthing people who were alone, or receiving bad news:

**Table 2** Theme and associated meaning units from the documentary and interview data

| Themes | Meaning units | |
| --- | --- | --- |
| | Documentary data | Interview data |
| Postcode lottery of care | Different policies used in local situations | Trust dictates rationale for decision-making |
| | Tensions between national and local policy and practise | Differences between trusts resulting in geographical variations |
| Confusion and stress around rules | Concern over transparency, clarity and rationale for decision-making | Confusion with rules leads to frustration |
| | | Confusions between staff about the rules |
| Unintended consequences | The need for companions as they improve well-being and outcomes for women/birthing people (and the negative impact of not having companions) | Lack of companionship created a distressing and frightening experience for women/birthing people |
| | The unintended consequences of lack of/ restrictions on companions | Lack of support for women/birthing people from companions |
| | The presence of companions supports staff | Increased work burden for staff |
| | The need to provide alternative support for women/birthing people if companion not present | Being alone when getting bad news at the scan |
| Need for flexibility | Need for consideration of women/birthing people who are identified to be particularly vulnerable, marginalised or need extra support (eg, due to ethnicity, language issues and baby loss) | Maternity services should be an exception |
| | The need to look at situations on a case-by-case basis to support personalised care | Rules should be applied flexibly to meet the needs of vulnerable women/birthing people |
| | | Importance of being flexible with rules when babies die |
| Acceptable time for support | The use of virtual means to replace physical companionship in antenatal scans | Ultrasound—rigidity and lack of flexibility |
| | Concerns over lack of companionship in early labour and for women/birthing people who are induced and need for support at this time | Knowing when to bring the companion in with active labour—impacts of being in prolonged labour alone |
| | Concerns over women/birthing people only allowed support in 'active' labour and how this is determined | Issues around companionship at the time of induction |
| Loss of human rights for gain in infection control | The balance between risk of transmission and the risks to women/birthing people | Expectation that women's/birthing people's rights around childbirth needed to be sacrificed for safety |
| | The assertion of women's/birthing people's (and companions) human rights as the basis of companionship | Limited focus on safety, centred on infection control |

All of a sudden it was just women on their own for us. And that was really, really stressful for the women and the staff. And a lot of a lot of my job is giving bad news. And to give that to women that are on their own with no support; you can't touch them. You can't hug them. And so that for us is really, really challenging. I think that was probably the most challenging thing. (Stakeholder 24, Sonographer)

### Need for flexibility

Concerns discussed within the documents (eg, BR_8; BR_18; RCM_4) and raised by stakeholders related to the blanket adoption of visitation rules across whole hospitals. Some commented that pregnant women/birthing people were a 'separate population with separate needs'

(Stakeholder 7, Service-user organisation), arguing that visiting rules adopted in other areas of healthcare should not apply to a perinatal population:

So, you know, several heads of midwifery were saying to me, I want to do this, but they won't let me because they made a decision about what the visiting will look like in this hospital. And they do not see maternity as an exception. And, you know, it is an interesting reflection, isn't it, that maternity has always been a service that has seen itself as an exception to the health care service in which it sits. (Stakeholder 17, NHS improvement lead)

Responses from organisations, including AIMS, Birthrights and the RCM, argued how there needed to be

consideration of women's/birthing people's unique situations. Restrictions on companionship were considered to have a disproportionate impact on those who were facing disadvantages, including those for whom English is a second language, those with mental health problems, cognitive impairments, refugee and asylum seekers (AIMS_8; RCM_4; BR_8; BR_18). Only one Trust included a statement on their website about offering personalised (flexible/individualised) care for all women/birthing people, that might not be in line with COVID-19 policies. Five others said that they offered this on a case-by-case basis (often expressed as 'exceptional circumstances'). One of these referred to Black, Asian and ethnic minority communities along with concerns about greater COVID-19 risks, and three referred to 'allowing' women/birthing people to bring a companion if they 'needed assistance'. In two Trusts, this was explicitly linked to those with disabilities:

> Partners and family members will not be allowed to enter the building unless you need support from a carer/relative (eg, if you have a disability and need support). (North-west 2)

Birthrights stipulated how 'protected characteristics' under the Equality Act 2010 (eg, physical disability or mental health condition) meant that maternity Trusts were obliged to make reasonable adjustments (eg, BR_15; BR_18). NHS England emphasised the need for 'essential visitors' (seen as different to 'normal visitors') for those with specific communication or care needs (NHSE_8). AIMS also stressed that on some occasions, on a 'case-by-case basis', such as a disability, 'a second birth partner may be critical to women's mental well-being or other needs' (AIMS_2).

The lack of flexibility for highly sensitive events such as experiences of trauma or loss were also highlighted by Birthrights (eg, BR_18) and the RCM (eg, RCM_41). While some incidents of positive practice were identified, stakeholders also shared stories of those whose baby had died in utero being unable to take photographs or spend time with their deceased child:

> We had a lot of stories from parents who hadn't been allowed to take photographs, haven't had time to sit and hold their baby. And I think all of those were linked to both a lack of space, to lack of bereavement space, but also a lack of staff understanding of how to adapt bereavement care standards. We also saw in this group a lot of problems around not having the partner with them. (Stakeholder 10, National charity)

### 'Acceptable' time for support

Access to and timing of support from companions were issues at key stages during the perinatal journey, and notably during antenatal ultrasound appointments and during early onset of labour. A key area of contention related to women/birthing people having to attend ultrasound scans unaccompanied: a situation described by one

of the stakeholders as 'ludicrous' (Stakeholder 7, Service-user organisation). Guidance from the Royal College of Obstetrics and Gynaecologists (RCOG) recommended that 'patients should be asked to attend alone if possible or with a maximum of one partner/visitor' (RCOG_5), whereas a RCM document stated 'partners should attend scans unless rooms are too small to socially distance: partners may attend scans virtually' (RCM_28). However, in contrast to this permission for virtual contact, a joint statement by the Society of Radiographers (SoR), RCOG, RCM and the British Medical Ultrasound Society stated that devices required for remote contact by companions via video/phones are a vehicle for transmission (due to surface contamination), and that recordings would impact on scan time, sonographer concentration and potential detection of fetal abnormalities—although it was acceptable for the woman/birthing person (if in line with local policies) to 'save a short 10–30 s cine clip of the fetus at the end of selected examinations' (SoR_11).

Trust data revealed that while most permitted companions at one or both standard ultrasound appointments (12 weeks and 20 weeks), seven (~30%) did not. Four Trusts mentioned video or other means of 'virtual' companionship, but usually to specify that videos of scans were not permitted. Only one Trust referred to women/birthing people being able to phone a companion for support if the sonographer were to find 'important clinical information that your partner needs to be aware of' (Greater London 2). While many stakeholders were critical of the ultrasound restrictions on the right to be accompanied, one respondent argued the need to highlight that antenatal ultrasound scanning continued, even when 'other screening programmes went into hibernation' (Stakeholder 26, Radiographer). Some defended restrictions on women/birthing people being accompanied during scanning, noting that scans often have to take place in areas 'like a broom cupboard in a very small poorly ventilated space' (Stakeholder 17, NHS improvement lead) coupled with the restricted time to undertake the examination and sterilising the room and equipment after each appointment. One reported that there had been a 'downgrading' of the importance of scans as a medical examination that required focused concentration in challenging situations, during vociferous debate about companion attendance (Stakeholder 26, Radiographer). However, while sonographers may have faced increased risks due to screening large numbers of women/birthing people, the specific rationale for not allowing videos as an alternative was challenged:

> You can argue the toss as to whether some of the justifications for not allowing that were real or weren't real. You know, is there really a risk of infection if you pick up your phone? Really? Maybe some anxiety for sonographers or whoever's doing the scan. You know, you don't really want the phone on with a video while you're doing the scan because who knows, they might use it in some kind of litigation. Who knows? But

whatever it was, it really didn't help. (Stakeholder 20, National charity)

A further area of contention concerned companionship during labour and birth. While organisations such as Birthrights argued for companionship throughout, less than half (9/25) of the Trusts referred to companions of choice attending 'throughout' or 'for the duration'. Three Trusts referred to companionship being permissible only when the woman/birthing person was in 'established' or 'active' labour (with no details as to how this would be established), and 13 Trusts did not specify the relevant phase of labour. RCM guidance advised that women/birthing people would not be able to have companions present during inductions that took place in a bay or ward (RCM_27). Only six Trusts (25%) indicated that companions could be present during induction of labour, and four allowed companions, but with limitations (either restricted to daytime or if the woman/birthing person needed additional support). About half (12) provided no information, and three explicitly stated that companions were not allowed:

> If you are attending for induction of labour please attend alone, your birth partner will join you once you are transferred to the Delivery Suite. (Yorkshire and Humber 1)

Birthrights and AIMS (ie, BR_8; AIMS_5; BR_18; BR_23) also raised concerns about cervical dilatation as the only acceptable indicator of active labour. As this meant that some women/birthing people who may not have wanted (or needed) a vaginal examination felt pressured to accept the procedure if they wanted their chosen companion to be granted access.

There were examples of innovation to try to support companionship. Some Trusts initiated or extended the provision of labour induction in community settings or in private hospital rooms (rather than multi-occupancy early labour wards) to prevent separation of women/birthing people from their companions (RCM_8; RCM_27). Some stakeholders also referred to more flexible approaches to induction such as companions being able to 'come and settle them [women] in' and to use 'Facetime to be with their partner all the time' (Stakeholder 21, Midwife—national role).

### Loss of human rights for gain in infection control

There was some evidence from stakeholders that hospital decision-makers in some settings believed that companionship during the maternity episode should not be prioritised over visiting rights in other areas where attendance of close family members would usually be seen as a critical human need and right: especially when someone was dying in hospital:

> So, it was interesting and when I would speak to the head of midwifery, sometimes it felt like they were saying, you know, well, everyone's got to make … sacrifices. And there are people dying alone in hospital.

There are people suffering terribly alone in hospital, unable to have visitors … . [while] there were women saying, you know, it's my right to have a companion it's your job to provide care for me. So, it felt at times like each group with their own concerns was unable to think about or found it difficult to take on board the concerns of the other group. (Stakeholder 17, NHS improvement lead)

The underlying principle within most of the guidance reviewed was that 'safety' was primarily conceptualised as the prevention of transmission of infection, for women/birthing people, companions and staff. NHS England documents referred to minimising 'control risks working with your IPC [Infection, Prevention & Control] leads, while still allowing the maximum possible safe access' (ie, NHSE_9). The RCOG/RCM also noted the need to minimise the number of attendees, but acknowledged that one person could be there for antenatal visits should a woman/birthing person choose this:

> You will be asked to come alone to clinical appointments or keep the number of people with you to one (including midwifery visits in your home). This will include being asked not to bring your children with you to appointments. This is important to protect maternity staff, other women and babies, and you and your family from the risk of infection. (RCOG/RCM_1g)

Birthrights was one of the key organisations to recommend that notions of safety might also include emotional and psychosocial risks of women/birthing people being unattended (eg, BR_17; BR_16; BR_19; BR_18)—'The damage caused by ongoing restrictions needs to be weighed up against the requirements of infection control' (BR_19). Several of their documents (eg, BR_18; BR_19) claimed that routinely restricting companions was a violation of women's/birthing people's (and companions) human rights. Despite this, stakeholders stated that, in practice, human rights and choices around companionship did not feature as part of the decision-making processes:

> And I think we've spent the last, you know, however, many years banging on about the fact we want to give women choice and rights and sharing that discourse and encouraging women to become empowered. And then COVID-19 comes along and we just say, no, no, we're not doing that. (Stakeholder 17, NHS improvement lead)

The RCM (RCM_41) stated that their 'greatest concern' was 'safety being sacrificed in favour of popularity', which seemed to imply that companionship or visiting should not outweigh the need to prevent infection of its 'members' and of 'women and families'. SoR also highlighted that its guidance had 'risk assessments' at its core (SoR_6). The guidance did not preclude 'people being accompanied', but that it 'must only happen if

the safety of the patient and sonographer is not compromised' (SoR_8). However, others argued that day to day decision-making was based around a belief about safety that was limited: 'because it's not just about the physical self, it's about [the] psychological self' (Stakeholder 14, Midwife—national role):

> But I think safety generally is an interesting thing because … You know, so many different things affect safety don't they, so something like being able to have your partner with you might not be seen as a primary thing affecting safety in comparison with protecting against COVID-19, but actually if it impacts on someone's mental health in either the partner or the mother, that does have an effect on safety. (Stakeholder 5, Maternity Voices Partnership)

## DISCUSSION

In this paper, we have drawn on guidance from national statutory and service-user organisations, key stakeholders and public-facing Trust-level data to consider the organisational issues associated with companionship and visiting in antenatal and intrapartum care during the COVID-19 pandemic. The terms companionship and visiting were not always clearly differentiated in data relating to the antenatal and intrapartum period, though most sources were consistent in referring to accompaniment as 'companionship' during labour and birth. The value of active companionship during labour and birth for women/birthing people is widely recognised, in terms of clinical benefits, and short-term and long-term psychosocial impacts.[1 19] As evidenced within this paper, during the COVID-19 pandemic, at the policy and organisational level, assumptions and norms about companionship, accompaniment and visiting during facility-based healthcare provision have faced profound challenges. Some of the key organisational challenges have concerned personnel shortages, infection control and restricted space. Others have noted the variance in maternity organisation response during the pandemic.[20] Some variation can probably be explained by changing national knowledge about the prevalence and impacts of COVID-19, and by different levels of exposure to COVID-19 infection. However, our data suggest that this was not the case where blanket policies were applied with minimal individual flexibility, or where there was unjustified variation in visiting and companionship rules, coupled with poor and inconsistent communication. There were no clear patterns in the Trust-level data that would allow us to explain the differences we documented in responses. While population level disparities may be a contributing factor, most Trusts serve a range of sociodemographic/economic areas, and other potentially relevant information such as space constraints was not publicly available.

We found particular concern about lack of access to companionship (in the sense of informational, practical and social support and advocacy[1]) in two distinct areas.

First, women/birthing people being unable to have any communication (actual or virtual) with companions at ultrasound scan; and, second, denial of intrapartum companionship until labour was 'established'. In relation to the former case, there is some evidence that, beyond the emotional and psychological benefits for the mother, when fathers and co-parents are present for antenatal ultrasound scan, there are significant effects on their identification with the fetus (as their future child) and their empathic relating with the woman/birthing person.[21 22] This implies that being present for ultrasound scans is more than simply 'visiting'. It has important public health and relationship benefits for the woman/birthing person, their partner and baby. In the latter case, in some Trusts, ensuring that labour had progressed sufficiently was perceived by some stakeholders to be associated with coercive and invasive practices, such as regular vaginal examinations when women/birthing people may otherwise not have needed or wanted such examinations. General uncertainty over organisational companionship permissions during labour and birth may also be reflected in anecdotal rises in women/birthing people choosing to freebirth,[23 24] and the associated RCM guidance to ensure appropriate professional responses.[25] Trust policies that restricted intrapartum companionship until labour was established (or until birth was imminent) seemed to be built on an assumption that companionship was only really needed when labour was very intense, and/or when the birth was happening, so that the companion could be 'permitted' to witness the birth of the baby. In contrast, other Trusts seemed to recognise, at the organisational level, that active and engaged companionship throughout labour (from the early stages of spontaneous labour, or from the time of labour induction through to the birth) is a mechanism for clinical, psychological and emotional safety for the woman/birthing person, partner and child, both in the short term, and, critically, in the longer term, when the threat of COVID-19 infection is long over.[12 26]

The pandemic brings into sharp focus the fundamental and underpinning ethical dilemma between social actions that ensure the greatest benefit for the population as a whole, and the individual human rights of each person within that population.[27] Resolving this potential conflict of ethical imperatives depends on an open and informed debate about rights and consequences. In terms of maternity care, this requires a sophisticated understanding of what 'companionship' (as opposed to 'visiting') means, over the whole life course, and for the woman/birthing person, partner, baby and family. It also requires attention to the potential moral distress of maternity care staff (and healthcare staff in general, including ultrasonographers). These professionals are faced with the stress of having to balance these two imperatives with real people, in intensely emotional real time, repeatedly day in and day out, and at times with insufficient PPE equipment available, at a time when they too could be pregnant at risk of exposure to infection, or fearful of infecting others.[28–31]

This is the first study to bring together national policy and organisational stakeholder views with Trust-based public-facing data to understand how companionship and visiting in antenatal and intrapartum care has been organised in England during COVID-19. Although we cannot be sure we captured every single relevant document produced over the period of our data collection, triangulation across data sources enabled rich insights into how and why variations occurred, and the perceived impacts. Returning quotes to stakeholders (as they requested), also provided a further level of rigour. The pragmatic restriction of the Trust-level data collection to only 25 Trusts (10% of maternity care providers in the UK), and the restriction to maternity-specific documents and guidance may be a limitation. However, the organisations that were included were selected purposively to reflect a wide range of relevant characteristics. Trust-level data were collected during a discrete period (September 2020 and October 2020), aiming to capture responses to changed national guidance; this limitation means we do not address how Trusts continued to respond to the changing pandemic. Since this paper is focused on policy and organisational responses to the pandemic, the views of women/birthing people, companions and healthcare professionals at Trust level were not included. In addition, our analysis did not include findings related to postnatal care, or care in neonatal units. These areas, and the unintended (positive and negative) short- and longer-term consequences of different interpretations of the value of companionship (in itself, and as opposed to being a hospital visitor), when balanced against infection control, are critical areas for examination during the on-going COVID-19 crisis. Future outputs from the ASPIRE project will address these gaps.

## CONCLUSION

This paper presents insights from the ASPIRE COVID-19 UK study to understand how companionship and visiting in maternity care was operationalised at the organisational level in antenatal and intrapartum care during COVID-19. Our findings illustrate variations in policy at national and local level, coupled with poor and inconsistent communication of how the restrictions changed in some sites, and a lack of clarity in the decision-making processes. The evidence highlights a lack of flexibility in responding to women/birthing people with more complex needs, the negative and positive unintended consequences of companionship restrictions, and the challenges of conceptualising and balancing infection risk and emotional and psychological distress. However, there was evidence that creative solutions were possible, since, despite significant pressures, some Trusts appeared to continue to provide full companionship.

Overall, these concerns illustrate something much more fundamental than merely barriers to hospital 'visiting'. While the NHS England Better Births policy agenda highlights the need for safety and personalisation within maternity care, these findings suggest that, over the time period captured by this study, personalisation (and emotional and psychological safety) became sacrificed in some (but not all) situations to the overriding imperative to minimise infection spread with high emotional and psychological costs. Further research should capture the views and experiences of healthcare professionals, women/birthing people and clinical outcome data from different settings. There is an urgent need to determine how to balance risks and benefits sensitively and flexibly and to create optimum outcomes for women/birthing people, companions (including fathers, co-parents and others), infants, families and staff, during the current and future crisis situations.

**Author affiliations**
[1]Maternal and Infant Nutrition & Nurture group, School of Community Health and Midwifery, University of Central Lancashire, Preston, UK
[2]Research in Childbirth and Health group, School of Community Health and Midwifery, University of Central Lancashire, Preston, UK
[3]North West Ambulance Service NHS Trust, Bolton, UK
[4]School of Medicine, University of Central Lancashire, Preston, UK

**Collaborators** ASPIRE-COVID-19 Collaborative Group: co-investigators: Soo Downe; University of Central Lancashire, George Ellison; University of Central Lancashire, Alan Fenton; Newcastle upon Tyne Hospitals NHS Foundation Trust, Alexander Heazell; University of Manchester, Ank de Jonge; Amsterdam University Medical Center, Carol Kingdon; University of Central Lancashire, Zoe Matthews; University of Southampton, Alexandra Severns; NHS England and NHS Improvement North West, Gill Thomson; University of Central Lancashire, AT; University of Central Lancashire, Alison Wright; Royal Free Teaching Hospital in London. Research staff: Naseerah Akooji; University of Central Lancashire, Marie-Clare Balaam; University of Central Lancashire, Jo Cull; University of Central Lancashire, Lauri van den Berg; Amsterdam University Medical Center, Nicola Crossland; University of Central Lancashire, Claire Feeley; University of Central Lancashire, Beata Franso; Amsterdam University Medical Center, Steph Heys; University of Central Lancashire, Gill Moncrieff; University of Central Lancashire, RN; University of Central Lancashire, AS; University of Central Lancashire. Steering committee: Maria Booker; Birthrights, Jane Sandall; Kings College London, Jim Thornton (chair); the University of Nottingham, Tisian Lynskey-Wilkie; University of Central Lancashire, Vanessa Wilson; Lancashire and South Cumbria Local Maternity System. Stakeholder group: Rebecca Abe and Tinuke Awe; FivexMore, Toyin Adeyinka; MVP BAME group, Ruth Bender-Atik; the Miscarriage Association, Lia Brigante; RCM, Rebecca Brione; Birthrights, Franka Cadée; International Confederation of Midwives, Elizabeth Duff; Expert, postnatal care, Tim Draycott; Royal College of Obstetricians and Gynaecologists, Duncan Fisher; fathers included/family included/the family initiative, Annie Francis; Independent Midwifery Advisor, Arie Franx; Erasmus MC, Lucy Frith; University of Liverpool, Louise Griew; National Maternity Voices, Clea Harmer; Sands, Caroline Homer; Burnet Institute, Australia, Marian Knight; National Perinatal Epidemiology Unit, Amali Lokugamage; Whittington Health NHS Trust/University College London, Amanda Mansfield; London Ambulance Service Trust, Neil Marlow; University College London, Trixie Mcaree; NHS England, David Monteith; Grace in Action, Keith Reed; Twins Trust, Yana Richens; UCL and City University, Lucia Rocca-Ihenacho; Midwifery Unit Network, Mary Ross-Davie; Royal College of Midwives Scotland, Seana Talbot; BirthWise NI, Myles Taylor; British Maternal and Fetal Medicine Society, Maureen Treadwell; Birth Trauma Association.

**Contributors** SD and AT with input from the ASPIRE-COVID19 Collaborative Group designed the study. RN, GT, SD, and NC interviewed the stakeholders; GT, MCB, GM and JC were involved in data extraction for the documentary analysis; SH, AS and NC collected and analysed Trust level data. GT, M-CB, and RN were involved in developing themes from the interview and documentary data, and GT synthesised the Trust level data into the data set. Final themes were agreed with all authors. All authors and the ASPIRE-COVID19 Collaborative Group contributed to writing and reviewing the manuscript. SD is guarantor for the study and the data contained within.

**Funding** This research is funded by the Economic and Social Research Council (ESRC), as part of UK Research and Innovation's rapid response to COVID-19 (grant number: ES/V004581/1). Full details of the main study are available via ResearchRegistry (researchregistry5911) and via UKRI Gateway (https://gtr.ukri.org/projects?ref=ES%2FV004581%2F1).

**Competing interests** None declared.

**Patient consent for publication** Not applicable.

**Ethics approval** This study involves human participants and was approved by health ethics subcommittee from the lead author's institution: University of Central Lancashire (project number: 0079). Participants gave informed consent to participate in the study before taking part. Participants were asked if they would like to check any public-facing quotes prior to publication as part of the consent process, with quotes returned to seven participants for feedback, and four required slight changes to be made. All feedback either involved minor grammatical changes; ensuring the quote was being used in the correct context, or, in one case, a change to job title.

**Provenance and peer review** Not commissioned; externally peer reviewed.

**Data availability statement** Data are available in a public, open access repository. Data are available upon reasonable request. Trust level data is included in the paper. Details of all documents analysed is provided/with all information freely available. All relevant interview data concerning companionship are available in UCLanData repository at https://doi.org/10.17030/uclan.data.00000319.

**ORCID iDs**
Gill Thomson http://orcid.org/0000-0003-3392-8182
Rebecca Nowland (Harris) http://orcid.org/0000-0003-4326-2425
Anastasia Topalidou http://orcid.org/0000-0003-0280-6801

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
