## [Reviewer comments · BMJ Open]

ARTICLE DETAILS

TITLE (PROVISIONAL)	Companionship for women/birthing people using antenatal and intrapartum care in England during COVID-19: A mixed-methods analysis of national and organisational responses and perspectives
AUTHORS	Thomson, Gillian; Balaam, Marie-Claire; Nowland (Harris), Rebecca; Crossland, Nicola; Moncrieff, Gill; Heys, Stephanie; Sarian, Arni; Cull, Joanne; Topalidou, Anastasia; Downe, Soo

VERSION 1 – REVIEW

REVIEWER	Dev, Alka Dartmouth Coll, The Dartmouth Institute
REVIEW RETURNED	16-Jul-2021

GENERAL COMMENTS	This is an important study and has implications for similar evaluations across the world as maternal health care access has suffered considerably under COVID 19. Minor recommendations: 1. Reword the objective (in abstract) to include "To explore the impact of COVID-19 policies on companionship for women using maternity services in England..." as it was unclear at first that the review would not include any interviews with women themselves. Another option is to also use the words 'companionship fulfillment' or 'companionship application' or 'companionship operationalization' rather than 'companionship' alone.2. On p4, line 48, possible to quantify 'liberal'? Are companions consistently allowed to accompany women for all visits? Are there restrictions on # of companions? Would be good to know the current norms.3. p.6 Under Methods, would be good to say 'This qualitative study is part of...'4. p.6 The development of themes needs more explanation - can you provide a table or illustration of how themes emerged from codes (i.e. what codes went into each theme). This is help to explain themes better (for example I really did not understand what was captured under postcode lottery theme exactly.5. For Table 1, is it possible to provide role for all? For example, stakeholder 4 is a midwife? Current table as it is does not communicate necessary information to anyone not in the know about these groups. Infact having a table of just roles with the number of stakeholders who fit that role would be more helpful for an international audience esp since organizations are listed again in Box 1. With alterations in accompanying text as needed.
---

	6. Table 3 is distracting. Can be deleted? See point 4 above about table regarding themes. Discussion: Are there known characteristics of Trusts that did better that could potentially explain why? For example, were wealthier postcodes more likely to be more permissive? In other words, can the variation in operationalization be explained through Trust level factors that reflect population level disparities or inequities? I am particularly thinking about Table 2 and Trust regions. 2.
--	---

REVIEWER	Berger, Blair Johns Hopkins University, Population, Family and Reproductive Health
REVIEW RETURNED	20-Jul-2021

GENERAL COMMENTS	Reviewer Comments: I extend my sincere thanks for the opportunity to review this manuscript on such a critical topic. Please see my detailed comments below by section of the paper. General Comments: The paper reads generally well, but an overall edit for grammatical errors, punctuation, writing clarity, and consistency in capitalizations (e.g. 'maternity trusts,' vs. 'maternity Trusts') is warranted. Abstract  • Please include the year of data collection with the month date ranges. Background  • There seems to be little clarity in this paper, beginning with the Background, about what exactly constitutes companionship; indeed, this may reflect the confusion and precarity companionship has in the policy environments being studied. The impression at times is that companionship is conflated with visitation—and sometimes a group of visitors, while at other times the stance is that a companion is a designated role in maternity care interactions; the selected participant quotes in the Results section sometimes add to the confusion about the perceived role of a companion. The authors' statements of reflexivity in the methods situate their own personal perspectives of companionship, but clarity and consistency in communicating this throughout the paper is limited. A clear and reasoned foundation and literature review covering the following would, I believe, improve rationale for the study as well as the organization:  o 1) Defining companionship. We have a working definition of labor companionship, as well as other companionship roles in other stages of the maternity continuum. A clear statement of these would help ground the reader, including who can take on this role (doula, partner, other family, friend), always grounded in the birthing person's choice—choice about whether to have one present, and choice about who the designated companion should be. Note that the first paragraph does not even mention the term companion/companionship. I also think noting differences between labor companionship and antenatal companionship is important, as the latter is recommended as a continuous presence throughout labor and delivery. The roles of a companion antenatal
---

vs. intrapartum, and the differing evidence we have around companionship in these time periods, would be particularly salient since many of your results skew toward antenatal companionship.

- o 2) Literature cited overview of the import and benefits of companionship found in previous research in aspects such as labor progression and intervention, technical and experiential components of quality of care, etc; the first citation is a Cochrane review of qualitative literature on labor companionship, a great resource and list of citations to expound and set foundation for the health import of companionship. The authors might also consider stating that labor companionship is included as a recommended practice in WHO's most recent intrapartum care guidelines, as well as a quality metric for WHO's improving quality of maternal/newborn care in facilities standards.
- o 3) The status of companionship in the UK context. The authors might reconsider restructuring this paragraph (p. 4 lines 31-49) to include recent estimates of how common companionship is, what some of the trends have been more recently, and they might consider removing some of the more historical information for space considerations. A clearer picture of recent companionship in the UK: who companions are, how common, etc. might be a more useful background context.

Methods

- In the description of sampling for Trusts, the authors mention area-level deprivation (p. 5, line 33). How was deprivation measured in this context? Similarly, they refer to "population type" in line 36 as a micro-level factor, but more clarity around what constitutes population type in this sampling scheme is warranted.
- Documentary Review:
 - o I'm a little unclear about how "key sources" were selected—it seems these were not drawn from the Trusts, but were another data source all together—is that accurate? If so, more clarity is needed on how these key sources were identified and ultimately selected for documentary review.
 - Trust-level facing communication
 - o The authors just refer to "data related to companionship" as the focus of extraction in this section. They specify from which online mediums extraction occurred, but not specifically what kinds of data. The authors might consider detailing what kind of data for specificity.
- Interviews
 - o Were the participants for IDIs purposively selected? It seems so from the text description, but an explicit statement of the selection process is needed.
 - o Please state the number of interviews conducted before saturation was reached at the end of the paragraph, and indicate how many Trusts they represented. I would suggest moving this up from the first line of the Results to this section.
 - o To aid in understanding of broad readership, the authors might state specifically that interviews were held via videoconferencing (using Microsoft Teams), if that is indeed the mode of data collection; alternatively, if the interviews were via voice calling only, please state as much.
- Ethical Approval
 - o The authors' decision to remit the IDI data back to the interviewee to review is an interesting one, even if an accepted practice in qualitative research. However, given that some changes to the data were made, at least a brief statement of the possible consequences of impacts of this kind of member checking

	should be addressed in the Discussion or limitations sections later on.  • Data Analysis  o Was any software used during either coding or data synthesis? Or were these analyses all done by hand? Please specify these analytic details. o Understood that the authors use a framework approach to synthesize the data, but it sounds like the IDIs were analyzed using a thematic analysis once the framework was in place, is that accurate? Or was it a content analysis as with the Trust-level data? Please state explicitly the analytic method used to analyze the interview data. o Related, explanation of the actual framework is quite limited in both Methods and Results. Was this framework generated de novo, or adapting an existing one? Results  o My general comment related to the Results is that while the Methods indicated a framework analysis was done to synthesize the triangulated data, but the Results are ultimately presented to the reader as a thematic analysis. If a framework was developed through analysis and during the synthesis, what were its components? I would suggest an overall reorganization of the results around the elements of this framework to provide a guide for the reader through this lengthy section, including how the themes highlighted are related within this framework. If the framework ultimately was not as salient, then it should be deemphasized in the analytic description. o A clearer summary of organizational elements around companionship may also strengthen this section so the takeaways are clear, be they personnel shortage issues, infection control and space, impacts on women and families, etc. o I think a clearer summary of 2 apparent areas emerging from this data is needed to underscore these findings: 1) no communication with partners/families during antenatal care, and 2) delay or denial of companions during intrapartum care, contingent on certain timing of labor progression. o I have a remaining confusion about timing as it relates to the implementation of these various policies. It would seem that phase of the pandemic response is critical to understanding the context of these policies and their perceived impacts. Did the implementation of elements of the policies change over time in some Trusts? Was the implementation segmented, or retracted in others? How did timing impact the main focus of this paper, organization of services re: companionship? Discussion/Conclusion:  o Given my comments about the preceding sections, and in particular the Methods and Results, I will reserve my notes about the Discussion at this time as addressing those comments may impact edits to the Discussion. Thank you again for the opportunity to review this manuscript, and I hope my suggestions enhance the strength of the reporting
--	---

VERSION 1 – AUTHOR RESPONSE

Reviewer 1	
1. Reword the objective (in abstract) to include "To explore the impact of COVID-19 on companionship for women using maternity services in England..." as it was unclear at first that the review would not include any interviews with women themselves. Another option is to also use the words 'companionship fulfillment' or 'companionship application' or 'companionship operationalization' rather than 'companionship' alone.	To ensure complete clarity, we have reworded this text to: To explore stakeholders' and national organisational perspectives on companionship for women using maternity services in England during COVID-19, as part of the ASPIRE COVID-19 study.
2. On p4, line 48, possible to quantify 'liberal'? Are companions consistently allowed to accompany women for all visits? Are there restrictions on # of companions? Would be good to know the current norms.	Additional information now added: "A survey published in 2013 found that over half of fathers attended at least one antenatal check, and that 'almost all' were present for ultrasound screening in pregnancy, and for labour(10). In 2019 97% of women in England said that their partner or someone else close to them was involved as much as they wanted them to be during labour and birth(11)."
3. p.6 Under Methods, would be good to say "This qualitative study is part of..."	Thank you – as not all the data are 'qualitative' (e.g. %ages of Trusts) we have changed the text to read 'This mixed-methods study'
4. p.6 The development of themes needs more explanation - can you provide a table or illustration of how themes emerged from codes (i.e. what codes went into each theme). This is help to explain themes better (for example I really did not understand what was captured under postcode lottery theme exactly.	Thank you for the comment – and in line with reviewer 2 feedback, we have updated the analysis section and included a new table (Table 2) to help illustrate theme development.
5. For Table 1, is it possible to provide role for all? For example, stakeholder 4 is a midwife? Current table as it is does not communicate necessary information to anyone not in the know about these groups. Infact having a table of just roles with the number of stakeholders who fit that role would be more helpful for an international audience esp since organizations are listed again in Box 1. With alterations in accompanying text as needed. *Comment from the Editor: As a general rule, we allow a maximum of two indirect identifiers in tables (e.g., age and sex). Therefore, please bear this in mind when making your revisions as we do not wish you to compromise the anonymity of the participants.	We have now included a summary of roles in the text (and the identifiers for included quotes are all now updated) and the table has now been removed: "All the twenty-six interview participants held a national and/or strategic role in midwifery (n=9), obstetrics (n=1), neonatology (n=1), anaesthesia (n=1), radiography/sonography (n=2) or as an NHS improvement lead (n=1). One was from a doula organisation, three were from the Maternity Voice Partnership (a NHS working group comprising lay members and professionals dedicated to improving maternity care), five were from national charities (focused on birth trauma, premature/sick infants, stillbirth, miscarriage, multiple births) and two service-user organisations that campaign and advocate for maternity care improvements."

6. Table 3 is distracting. Can be deleted? See point 4 above about table regarding themes.	We feel this is an important table to demonstrate transparency and rigour but have now included this as a Supplementary File (File 3).
Discussion: Are there known characteristics of Trusts that did better that could potentially explain why? For example, were wealthier postcodes more likely to be more permissive? In other words, can the variation in operationalization be explained through Trust level factors that reflect population level disparities or inequities? I am particularly thinking about Table 2 and Trust regions.	This is an important question, but our reading of the data was that there were no clear patterns. While population level disparities may be a contributing factor, most Trusts serve a range of sociodemographic/economic areas, making it difficult to discern how this impacts and/or intersects with other variables such as varying prevalence of COVID-19 around England (which itself relates to socioeconomic variation), or characteristics not available in public-facing data such as Trusts' estate footprints or staffing levels. To reflect this complexity, we have added the following sentence in the discussion: "There were no clear patterns in the Trust-level data that would allow us to explain the differences we documented in responses. While population level disparities may be a contributing factor, most Trusts serve a range of sociodemographic/economic areas, and other potentially relevant information such as space constraints was not publicly available."
Reviewer 2	
The paper reads generally well, but an overall edit for grammatical errors, punctuation, writing clarity, and consistency in capitalizations (e.g. 'maternity trusts,' vs. 'maternity Trusts') is warranted.	Thank you, a careful proof check has been undertaken.
Abstract	
Please include the year of data collection with the month date ranges.	Now included
Background	
There seems to be little clarity in this paper, beginning with the Background, about what exactly constitutes companionship; indeed, this may reflect the confusion and precarity companionship has in the policy environments being studied. The impression at times is that companionship is conflated with visitation—and sometimes a group of visitors, while at other times the stance is that a companion is a designated role in maternity care interactions; the selected participant quotes in the Results section sometimes add to the confusion about the perceived role of a companion. The authors' statements of reflexivity in the methods situate their own personal perspectives of companionship, but clarity and consistency in communicating this throughout the paper is limited. A clear and reasoned foundation and literature review covering the following would, I believe, improve rationale for the study as well as the organization:	Thank you – we do appreciate the importance of the literature on the nature of companionship, and, indeed, we have previously published a review of the literature that concluded the status of the birth partner was 'not-patient, not-visitor'. We had originally considered a discussion of the difference between companionship and visitor status for this paper, but did not add this in to the submitted version as we felt it wasn't central to the focus of our paper. We have now added some text throughout the introduction and the discussion to encompass the key points in these four comments. However, because the definition of companionship per se was not the focus of this particular analysis, we have not gone into this topic in great depth in the current paper, or added a long list of references – though we have added the Cochrane review on continuous support in labour as we agree that this is missing. We have now, therefore, referenced the two key qualitative and quantitative systematic reviews in this area.
1) Defining companionship. We have a working definition of labor companionship, as well as other companionship roles in other	The reviewer comments have prompted us to consider if there is another paper to be written that takes a more semantic approach to analysing how companionship is

stages of the maternity continuum. A clear statement of these would help ground the reader, including who can take on this role (doula, partner, other family, friend), always grounded in the birthing person’s choice—choice about whether to have one present, and choice about who the designated companion should be. Note that the first paragraph does not even mention the term companion/companionship. I also think noting differences between labor companionship and antenatal companionship is important, as the latter is recommended as a continuous presence throughout labor and delivery. The roles of a companion antenatal vs. intrapartum, and the differing evidence we have around companionship in these time periods, would be particularly salient since many of your results skew toward antenatal companionship.	framed and interpreted across the data we have collected for this paper, and for the wider study (that includes interviews at Trust level with women and staff). Thank you to the reviewer for raising this possibility among the research team We are rather surprised that the reviewer states that our results skew towards antenatal companionship. We have checked the paper, and feel that labour and birth companionship are represented at various points in the paper, as dictated by our data. Our discussion and conclusion covers both periods. Where it may have been less clear, we have now specified when we are talking about labour and birth companionship.
2) Literature cited overview of the import and benefits of companionship found in previous research in aspects such as labor progression and intervention, technical and experiential components of quality of care, etc; the first citation is a Cochrane review of qualitative literature on labor companionship, a great resource and list of citations to expound and set foundation for the health import of companionship. The authors might also consider stating that labor companionship is included as a recommended practice in WHO’s most recent intrapartum care guidelines, as well as a quality metric for WHO’s improving quality of maternal/newborn care in facilities standards.	We have made numerous tweaks to emphasise that the paper concerns companionship/visiting during the antenatal and intrapartum period, i.e. title, objective and elsewhere. We have also noted the lack of focus on postnatal care within the strengths and limitations. We intended to write a paper that focuses on postnatal (and to cover neonatal) care from our case study data collection. Cochrane review on continuous support in labour now added. The WHO recommendation for companionship (2016) that was cited in the original paper (ref 9) is now replaced with the most recent 2018 recommendations. Additional information now added: A survey published in 2013 found that over half of fathers attended at least one antenatal check, and that ‘almost all’ were present for ultrasound screening in pregnancy, and for labour(10). In 2019 97% of women in England said that their partner or someone else close to them was involved as much as they wanted them to be during labour and birth(11).
3) The status of companionship in the UK context. The authors might reconsider restructuring this paragraph (p. 4 lines 31-49) to include recent estimates of how common companionship is, what some of the trends have been more recently, and they might consider removing some of the more historical information for space considerations. A clearer picture of recent companionship in the UK: who companions are, how common, etc. might be a more useful background context.	
Methods	
In the description of sampling for Trusts, the authors mention area-level deprivation (p. 5, line 33). How was deprivation measured in this context? Similarly, they refer to “population type” in line 36 as a micro-level factor, but more clarity around what constitutes population type in this sampling scheme is warranted.	We used the English Indices of Deprivation (2019) as our measure of deprivation – we have clarified this in the Methods. We agree that our previous wording relating to micro level factors was unclear, so we have amended this to read “micro-level factors including aspects such as parity / access to care identified within national documents.”
Documentary Review:	
I’m a little unclear about how “key sources” were selected—it seems these were not drawn from the Trusts, but were another data	The documents are from all the organisations detailed in Box 1. We have added additional text to make this more explicit.

source all together—is that accurate? If so, more clarity is needed on how these key sources were identified and ultimately selected for documentary review.	“Guidelines, position papers, and reports relating to maternity care were collected prospectively between February and December 2020 from key governmental, professional, and service-user organisations. The organisations were identified as those who provide guidance, campaign and/or advocate for national practice and policy in relation to maternity care. The documents were sourced via organisation-based websites and from key stakeholders involved in the ASPIRE study. All documents that concerned maternity care provision were reviewed with any/all data in relation to companionship and visiting during antenatal and intrapartum care extracted and logged in Excel files.”
Trust-level facing communication: The authors just refer to “data related to companionship” as the focus of extraction in this section. They specify from which online mediums extraction occurred, but not specifically what kinds of data. The authors might consider detailing what kind of data for specificity.	We have added additional information to this effect in the Methods. “Data related to companionship and visiting in pregnancy, labour and birth were extracted from Trust websites and Trust-related Twitter, Facebook, and Instagram feeds, between September and October 2020. We extracted information on the format of information presented, access to companionship or visiting antenatally (including ultrasound), and during labour and birth (including induction of labour). We also extracted information that discussed personalisation, organizational response to specific additional needs, Trust response to national guidance, rationale for decisions about companionship/visiting rules, and any additional information on companionship/visiting in the context of COVID-19.”
Interviews Were the participants for IDIs purposively selected? It seems so from the text description, but an explicit statement of the selection process is needed.	Thank you – additional text has now been added. Purposive sampling was used to recruit individuals from relevant national governmental, professional, and service-user organisation leads involved in maternity care. Key individuals were identified by project and advisory teams and via snowballing.
Please state the number of interviews conducted before saturation was reached at the end of the paragraph, and indicate how many Trusts they represented. I would suggest moving this up from the first line of the Results to this section.	Saturation is a rather contested concept (e.g. Saturation in qualitative research: exploring its conceptualization and operationalization SpringerLink. We were more mindful of collecting the views from key stakeholders involved/concerned with maternity care. The individuals were not recruited based on whether they worked at a specific Trust. They had a more national, strategic, oversight of maternity care and policy. We have included additional text to make this more explicit. It is also not usual to include participant numbers in the methods – so whilst this has been moved, we would appreciate editorial advice on this. “A total of 50 individuals were approached to participate and interviews were held with twenty-six stakeholders. While some stakeholders did not respond to the requests, others provided names of individuals who they considered would be more suitable. Recruitment was not based on saturation but was designed to ensure that we included

	representation from all key organisations and from individuals that were considered to offer important insights into maternity care delivery.”
To aid in understanding of broad readership, the authors might state specifically that interviews were held via videoconferencing (using Microsoft Teams), if that is indeed the mode of data collection; alternatively, if the interviews were via voice calling only, please state as much.	We have amended the text as detailed. “Semi-structured interviews were held July-December 2020 via videoconferencing (using Microsoft Teams).”
Ethical Approval The authors’ decision to remit the IDI data back to the interviewee to review is an interesting one, even if an accepted practice in qualitative research. However, given that some changes to the data were made, at least a brief statement of the possible consequences of impacts of this kind of member checking should be addressed in the Discussion or limitations sections later on.	We have added additional text as to what the required changes where (and in line with requested editorial changes): “All feedback either involved minor grammatical changes; ensuring the quote was being used in the correct context; or, in one case, a change to job title.” These were minimal tweaks that did not alter the meaning. We have also included some additional text in the discussion. Returning quotes to stakeholders (as they requested), also provided a further level of rigour.
Data Analysis Was any software used during either coding or data synthesis? Or were these analyses all done by hand? Please specify these analytic details.	Excel was used – this has now been included.
Understood that the authors use a framework approach to synthesize the data, but it sounds like the IDIs were analyzed used a thematic analysis once the framework was in place, is that accurate? Or was it a content analysis as with the Trust-level data? Please state explicitly the analytic method used to analyze the interview data.	Thank you – appreciate this was a little confusing, and this section has now been completely re-written. We have also included a table to demonstrate how the codes (meaning units) from the interview and documents informed the themes.
Related, explanation of the actual framework is quite limited in both Methods and Results. Was this framework generated de novo, or adapting an existing one?	Please see response above.
Results	
My general comment related to the Results is that while the Methods indicated a framework analysis was done to synthesize the triangulated data, but the Results are ultimately presented to the reader as a thematic analysis. If a framework was developed through analysis and during the synthesis, what were its components? I would suggest an overall reorganization of the results around the elements of this framework to provide a guide for the reader through this lengthy section, including how the themes highlighted are related within this framework. If the framework ultimately was not as salient, then it should be deemphasized in the analytic description.	Please see response above. We have totally re-written this section and included a separate table (Table 2) that shows how the codes from the documentary and interview data were collapsed into themes – with the Trust level data then synthesised at a later point. We hope this is clearer.

A clearer summary of organizational elements around companionship may also strengthen this section so the takeaways are clear, be they personnel shortage issues, infection control and space, impacts on women and families, etc.	Many thanks – we have included additional text in the discussion to emphasise these points: “Some of the key organisational challenges have concerned personnel shortage issues, infection control and restricted space.”.
I think a clearer summary of 2 apparent areas emerging from this data is needed to underscore these findings: 1) no communication with partners/families during antenatal care, and 2) delay or denial of companions during intrapartum care, contingent on certain timing of labor progression.	Many thanks for this point. These are areas that we already/had focused on in the discussion.
I have a remaining confusion about timing as it relates to the implementation of these various policies. It would seem that phase of the pandemic response is critical to understanding the context of these policies and their perceived impacts. Did the implementation of elements of the policies change over time in some Trusts? Was the implementation segmented, or retracted in others? How did timing impact the main focus of this paper, organization of services re: companionship?	Data from Trust public-facing communications represents a ‘snapshot’ of information logged between September and October 2020, following the publication on 8th September 2020 of NHS England guidance on companionship in maternity services, so does not address the question of responses over time. We have added a sentence to the limitations section of the Discussion as follows: “Trust-level data was collected during a discrete period of time (September and October 2020), aiming to capture responses to changed national guidance; this limitation means we do not address how Trusts continued to respond to the changing pandemic.”

VERSION 2 – REVIEW

REVIEWER	Dev, Alka Dartmouth Coll, The Dartmouth Institute
REVIEW RETURNED	12-Oct-2021
GENERAL COMMENTS	No additional comments.
REVIEWER	Berger, Blair Johns Hopkins University, Population, Family and Reproductive Health
REVIEW RETURNED	05-Oct-2021
GENERAL COMMENTS	Many thanks for the opportunity to review this revision. The content and organization of this draft has been strengthened significantly from the initial submission. Please see a couple of minor comments I offer for the authors' consideration to strengthen further: General: One more editing pass through the paper should help catch a few remaining grammatical issues. A global comment that occurs to me reading the reflexivity statements and Background, Discussion sections is the authors might consider clarifying that

	their orientation to labor companionship is prioritizing *choice* of the pregnant/birthing person. As the authors are familiar, norms around companionship differ widely between settings, communities, and individuals. It might help just to clarify that the priority is for policies to be explicit about supporting and facilitating companionship, if desired, rather than companionship will be a given for all individuals if policies allowed as a given. I also suggest some uniformity in how authors' refer to birthing/pregnant people and companions in the paper so it is abundantly clear when they are referring to a patient vs. a partner/companion. For example, the authors sometimes refer to "parents," "patients," "women" "birthing people" "partners", etc. When not a direct quote from a TRUST, the authors might consider uniformly using "patient" and "companion" for clarity and gender and role (i.e., not all birthing people will be parents, etc) neutrality. Background:  • Following Reviewer 1's previous comments on the original submission—hard to tell from the statistics provided in the resubmission on increases in companionship in the UK whether companionship has just become more popular, or whether these increases are due to policy changes that have become more inclusive • P. 4 line 45-46: The authors might consider revising for a clearer, stronger opening sentence that just states continuous labor companionship is an evidence-based practice with documented benefits and care experiences and clinical outcomes. • P. 4, line 50-51: Phrasing is unclear for the sentence on partner investment; please consider revising to make the point clear. I think the following sentence about differentiating between "hospital visitors" is not needed here; that distinction is made clearly and consistently throughout the paper; it just reads as extra and a bit clunky here. Methods  • Data analysis: p. 7 line 34/35: There may be a typo here, as the authors indicate five key themes were combined during analysis; all other parts of paper indicate 6 themes.
--	--

VERSION 2 – AUTHOR RESPONSE

Response to reviewers:

Many thanks for the opportunity to revise the paper – we are very grateful for the feedback received.

Comments to the Author:

Many thanks for the opportunity to review this revision. The content and organization of this draft has been strengthened significantly from the initial submission. Please see a couple of minor comments I offer for the authors' consideration to strengthen further:

Response: Many thanks for your support of our work

General: One more editing pass through the paper should help catch a few remaining grammatical issues. A global comment that occurs to me reading the reflexivity statements and Background, Discussion sections is the authors might consider clarifying that their orientation to labor companionship is prioritizing *choice* of the pregnant/birthing person. As the authors are familiar, norms around companionship differ widely between settings, communities, and individuals. It might help just to clarify that the priority is for policies to be explicit about supporting and facilitating companionship, if desired, rather than companionship will be a given for all individuals if policies allowed as a given.

Response: Thank you we have added some additional text at different points in the paper, i.e. introduction - Qualitative studies show that most women/birthing people value companionship, and global guidelines strongly emphasize the need to support and facilitate women's choice to be accompanied throughout the maternity journey (9).

Reflexivity statement - All authors believe that women/birthing people highly value companionship during key moments in their maternity care, that the priority for policies should be for supporting and facilitating companionship, if desired.

I also suggest some uniformity in how authors' refer to birthing/pregnant people and companions in the paper so it is abundantly clear when they are referring to a patient vs. a partner/companion. For example, the authors sometimes refer to "parents," "patients," "women" "birthing people" "partners", etc. When not a direct quote from a TRUST, the authors might consider uniformly using "patient" and "companion" for clarity and gender and role (i.e., not all birthing people will be parents, etc) neutrality.

Response: Thank you for these points, we have made some changes for consistency, but as patient can be associated with passivity, and more aligned with a general hospital population, we have included the terms women/birthing people for inclusivity and changed 'partners/parents' to companions.

Background:

- Following Reviewer 1's previous comments on the original submission—hard to tell from the statistics provided in the resubmission on increases in companionship in the UK whether companionship has just become more popular, or whether these increases are due to policy changes that have become more inclusive

Response: We have included another sentence here to address this point - This is likely related to more inclusive policies as well as consumer demand.

- P. 4 line 45-46: The authors might consider revising for a clearer, stronger opening sentence that just states continuous labor companionship is an evidence-based practice with documented benefits and care experiences and clinical outcomes.

Response: Thank you – this has now been included to read:

Companionship in maternity care is an evidence-based practice with documented benefits and care experiences and clinical outcomes (1) and has been associated with four key attributes: informational support, advocacy, practical support, and emotional support(1)

- P. 4, line 50-51: Phrasing is unclear for the sentence on partner investment; please consider revising to make the point clear. I think the following sentence about differentiating between "hospital visitors" is not needed here; that distinction is made clearly and consistently throughout the paper; it just reads as extra and a bit clunky here.

Response: Thank you – we agree with your points, and have now removed both sentences as felt to be unnecessary from the central arguments.

Methods

- Data analysis: p. 7 line 34/35: There may be a typo here, as the authors indicate five key themes were combined during analysis; all other parts of paper indicate 6 themes.

Response: Many thanks for spotting this error – now revised to six.